# Synthesis and Characterisation of Graphene Oxide-Silica-Chitosan for Eliminating the Pb(II) from Aqueous Solution

**DOI:** 10.3390/polym12091922

**Published:** 2020-08-26

**Authors:** Sepehr Azizkhani, Ebrahim Mahmoudi, Norhafizah Abdullah, Mohd Halim Shah Ismail, Abdul Wahab Mohammad, Siti Aslina Hussain

**Affiliations:** 1Department of Chemical and Environmental Engineering, Faculty of Engineering, Universiti Putra Malaysia, Serdang 43400, Selangor, DE, Malaysia; nhafizah@upm.edu.my (N.A.); mshalim@upm.edu.my (M.H.S.I.); 2Department of Chemical and Process Engineering, Faculty of Engineering and Built Environment, Universiti Kebangsaan Malaysia, UKM Bangi 43600, Selangor, DE, Malaysia; ebi.dream@gmail.com (E.M.); awm.ukm@gmail.com (A.W.M.); 3Centre for Sustainable Process Technology (CESPRO), Faculty of Engineering and Built Environment, Universiti Kebangsaan Malaysia, Kuala Lumpur 43600, Malaysia

**Keywords:** adsorption, graphene oxide, chitosan, synthesis, isotherm, adsorbent

## Abstract

Heavy metal ions have a toxic and negative influences on the environment and human health even at low concentrations and need to be removed from wastewater. Chitosan and graphene oxide are suitable nano plate adsorbents with high adsorption potential because of their π-π interaction, and they are available functional groups that interact with other elements. In this study, graphene oxide was coated with silica to enhance the hydrophilicity of the adsorbent. Subsequently, the adsorbent was functionalised by various amounts of chitosan to improve the Pb(II) removal. The adsorbent was analysed using transmission electron microscopy (TEM), Raman, Fourier-transform infrared spectroscopy (FT-IR), scanning electron microscope (SEM), and mapping analysis techniques. An investigation of the influences of the initial concentration of Pb(II), pH and contact time were included to obtain the optimum amount of adsorption. The range of the initial Pb(II) concentration studied was from 10 to 120 mg/L. The pH factor ranged from 3 to 8 with contact time from 0 to 140 min. Freundlich, Temkin and Langmuir isotherm models were fit to the results, and a pseudo-second-order kinetic model was found to provide a good fit as well. The maximum Pb(II) removal capacity achieved was 256.41 (+/− 4%) mg/g based on Langmuir isotherms.

## 1. Introduction

Water is known as a vital resource for life; nevertheless, numerous people around the world do not have access to clean and pure drinking water. Increasing population, unorganised industrialisation and the rapid rate of urbanisation lead to severe water contamination [1,2]. Releases of poisonous and untreated manufacturing effluents are the principal sources of contamination. Heavy metal ion pollution is one of several water pollution sources of concern that has a high negative influence on human health and the environment due to its toxicity [3], and the fact that it is non-biodegradable and has a tendency to accumulate [4,5]. Pb(II) is a dangerous heavy metal ion, which continues to be in broad use around the world for the foreseeable future, and which can be discharged into the environment from both natural activities and artificial processes. Pb(II) can have a negative impact on the nervous system, and can cause encephalopathy, cancer, renal kidney syndrome, nephritic syndrome, anaemia, hepatitis and intellectual disability in humans [6,7]. There are several sources of water pollution by Pb(II), such as printing methods, battery production, petroleum industries, plumbing, painting, metal plating and mining [6]. Pb(II) has a serious effect and causes notable damage to human health, even at low concentrations; the permissible Pb(II) amount in pure water is 0.01 mg/L, according to the World Health Organization. Consequently, the elimination of Pb(II) from wastewater before it is released into the ecosystem is important [8]. There are various techniques for eliminating heavy metal ions, such as electrochemical modification [9], precipitation [10], the membrane method [11], ion exchange [12] and adsorption [13]. Appropriately, adsorption has been identified as a more efficient and economical process in order to eliminate contamination from wastewater [13], and the selection of a good adsorbent is important for the Pb(II) adsorbing process.

Finding the suitable materials with various functional groups and interaction ability is the important issue in preparing the adsorbent. Graphene oxide (GO) has attracted significant attention as a promising alternative adsorbent material due to its special properties, such as its large surface area [14], chemical stability and remarkable mechanical durability [15]. It has been shown to have a high adsorption potential for the elimination of Pb(II), Cd(II) and Co(II) from aqueous solutions [6]. GO has chemically reactive oxygen functionality on the basal layers (hydroxyl and epoxy groups) and their edges (carboxylic acid), which increases reactions with other materials [16,17]. Silica contains numerous silanol groups (Si-OH) on its outer shell, which act as suitable nucleation and anchor points for the natural functionalisation of silica [18]. The surface of graphene oxide was coated by silica to improve the hydrophilicity features of composite [19]. Another material that has been studied in this study was Chitosan, with functional groups of -OH and -NH_2_ [20]. Chitosan can interact with heavy metal ions by chelation or ion exchange, because of its functional groups (amino and hydroxyl groups) [21]. 

The purpose of this study is to investigate a graphene-silica-chitosan (GOSCh) adsorbent with a high adsorption capability and superior separation features. Our hypothesis was that the decoration of chitosan and silica will enhance the adsorption capability, by adding different mechanisms of adsorption (Pi-, Pi and ion-exchange) and enhancing the adsorption capacity and resignation capability. The suitable adsorbent needs the various amounts of functional groups to interact with heavy metal ions and make a proper situation to interact with heavy metal ions effortlessly. Hence, in this study, for the novelty of this study, the silica was coated on the surface of GO and the combined with chitosan. Silica was used to increase the hydrophilicity of adsorbent with silane groups and let the adsorbent move inside the aqueous solution easily with reducing the resistance layer between the liquid solution and solid phase of adsorbent. Graphene oxide-silica and chitosan were combined to increase the adsorbent efficiency by various functional groups of GO and chitosan; increase the hydrophilicity by using silica, ion exchanging interaction between Pb(II) and the functional groups of adsorbent and synergic effect of chitosan and graphene oxide interaction. The results show significant enhancement toward the removal of Pb(II). In this new adsorbent, the zeta potential is negative, because of silica particles that undergo ion exchanging between positive and negative charges. Additionally, the multi-functional groups and π–π interaction of synthesised adsorbent increase the removal percentage pf Pb(II). We investigated the isothermal, kinetic and thermodynamic mechanisms of Pb(II) adsorption onto the GOSCh adsorbent. The results show that GOSCh can be applied as an effective adsorbent for removing Pb(II) from aqueous solution.

## 2. Materials and Method

### 2.1. Materials

Extra-pure, fine, natural graphite powder was obtained from Merck Co Malaysia (Selangor, Malaysia). Potassium permanganate (KMnO_4_) and sulfuric acid (H_2_SO_4_) (98 wt%) were supplied by Accot Malaysia. Chitosan (medium molecular weight) (Selangor, Malaysia), Pb(II) nitrate, and tetraethyl orthosilicate (TEOS) 98% were purchased from Sigma Aldrich Malaysia (Selangor, Malaysia). 

### 2.2. Preparation of Graphene Oxide-Silica-Chitosan

Natural graphite powder was used to prepare the GO using Hummers method [22]. Natural graphite (10 g) was added to sodium nitrate (5 g) of sodium nitrate, then put in the beaker with 230 mL of 98% concentrated sulphuric acid. After mixing all of these items for 2 h, the potassium permanganate added to beaker and the temperature of the beaker was controlled by ice bath and kept under 10 °C and stirred for 2 h. Then, the mixture was diluted with 500 mL deionised water slowly, and 20 mL of 30% H_2_O_2_ was added to oxidised the graphene, and then sent to the freeze dryer. TEOS was used for coating the GO surface with silica particles by in situ hydrolysis [19] in order to produce the graphene oxide-silica (GO-SiO). A total of 1 g of graphene oxide dispersed in the 5 L of ethyl alcohol and 1 L water and sonicated for 2 h. By adding the ammonia solution, the pH reached 9 and 1 mL of TEOS was added to the solution and sonicated for 1hr, then stirred for 1 day. The solution was washed 5 times by deionised water and centrifuged. Quantities of chitosan (0.2, 0.4, 0.6 and 0.8 g) were separately suspended in 10 mL of 1% (*v*/*v*) acetic acid solution for each 1 g of GO-SiO and stirred for 8 h, after which 1 g of GO-SiO was added to each chitosan solution and magnetically stirred for about 10 min to ensure homogeneity. Then, the mixture was slowly heated up to 50 °C and stirred for 3 h; the resulting mixtures were washed with acetone three times, filtered, and dried in a freeze dryer. The characterisation of GOSCh was analysed by using Fourier-transform infrared spectroscopy (FT-IR) (Spectrum ASCII PEDS 1.60, Waltham, MA, USA), transmission electron microscopy (TEM) (JEM-2100F electron microscope machine, company of JEOL, Tokyo, Japan), scanning electron microscope (SEM) (S-3400N, Hitachi-science & Technology, Krefeld, Germany), Raman (Witec Raman Microscope Machine, Ulm, Germany) and Zeta sizer (Nano-ZS, Malvern Instruments, Worcestershire, UK). The functional groups and elements were detected by FT-IR, the surface analysis, morphology and elements was studied using SEM, mapping, TEM and Raman.

### 2.3. Adsorption Process

The effects of significant factors such as the adsorbent dosage, initial concentration, pH and contact time were evaluated. GOSCh doses from 0.04 to 0.9 g were investigated. The initial concentrations of the Pb(II) solutions were in the range of 10 to 120 ppm. The influence of pH on the adsorption of Pb(II) from aqueous solution by GOSCh was investigated in different solutions, with a pH ranging from 3 to 8. Subsequently, the impact of contact time in optimum conditions was studied by varying the contact time from 5 to 140 min. After adsorbing the Pb(II) in the solution by GOSCh adsorbent, the samples were centrifuged and passed from the filter to separate the occupied adsorbent from the final solution. The concentration of solution after adsorption was decreased and set on the 5 ppm concentration which is the maximum standard of atomic absorption spectroscopy for eliminating the heavy metal ions. The adsorption capacity and adsorption percentage of Pb(II) by the GOSCh adsorbent were calculated with two equations: (1)$$q_{e}=\frac{\left(C_{0}-C_{e}\right)V}{M}.\;$$
(2)$$q\%=\frac{\left(C_{0}-C_{e}\right)}{C_{0}}\times 100\%.\;$$
where q is the adsorption capacity, *C*_0_ is the initial solution concentration, *C_e_* is the concentration solution after adsorption, V is the solution volume and *M* is the adsorbent mass.

## 3. Results and Discussion 

### 3.1. Analysis and Characterisation of Graphene Oxide-Silica-Chitosan Adsorbent

#### 3.1.1. FT-IR Analysis

FT-IR spectra of the GO and GOSCh are illustrated in Figure 1. The results indicated that natural graphite was strongly oxidised to graphene oxide, hence, C=O bonds appeared, represented by the peaks at 1736 cm^−1^ that correspond to the stretching vibration of C=O functional group [23]. Moreover, the spectra at 1067 and 1223 cm^−1^ represent the epoxy and alkoxy groups and bands corresponding to the C–O stretching vibrations [24]. Additionally, the C–O stretching vibrations of carbonyl and carboxylic acid groups are observed on the GO surface at 1665 cm^−1^. Additionally, the broad peaks around 3000 and 3609 cm^−1^ correspond to the stretching vibration and deformation vibration of O–H functional groups, respectively [8]. There are some peaks at 1066.94, 1223, 1665, 1736, 3609 cm^−1^ in the GOSCh FT-IR, and some new peaks or intensity change of peaks were shown. The GOSCh composite had firm peaks around 3546 cm^−1^ because of the O–H stretching vibration, N–H expansion vibration and intermolecular polysaccharide hydrogen bonds [25], although the specific peaks at 1622 cm^−1^ are related to the amide I (NH–CO) C=O stretching vibration, and the peaks at 1543 cm^−1^ are explained by the N-H bending of NH_2_ [25]. The peaks at 1367 cm^−1^ and 1066 cm^−1^ correspond to the C–OH stretching vibrations and C–O–C, respectively. In the GOSCh, the sharp peaks at 3542 cm^−1^ (O–H) and 1066 cm^−1^ (C–O) were higher than the same peaks in GO, because of the covalent interactions of OH groups with functional groups of chitosan. The peak at 1736 cm^−1^ moved to a higher frequency, demonstrating the reaction of chitosan with carboxylic groups and the deformation of carboxylic groups of graphene because chitosan functional groups (NH_2_) are able to react with oxygen groups of GO [26]. The peak at 696 cm^−1^ corresponds to the symmetric stretching deformation of silica groups, and the peak around 3425 cm^−1^ may be recognised as representing stretching of the O-H bonds of silanol groups [27]. Then, the silane groups of adsorbent caused an enhancement of the hydrophilicity, and decreased the resistance between the liquid solution and the outside surface of solid adsorbent. Furthermore, the functional groups of amine, amide, hydroxyl, carboxyl, ketone and epoxy of GOSCh, which have a negative charge, make a bond and interact with Pb(II).

#### 3.1.2. TEM Analysis

The morphologies exhibited by the GOSCh were imaged by TEM pictures, as shown in Figure 2. Figure 2a shows the GO sheet, and, in comparison with Figure 2b, it can be confirmed that the nanoparticles of silica were spread on the GO layers. In addition, Figure 2c shows the chitosan particles that were mixed with the silica coating on the GO-SiO. TEM analysis was provided, the changes are made on the surface of graphene oxide sheets are dramatic and clearly visible. Additionally, similar results were supported using the SEM mapping, which shows the presence of elements such as Si, N, C and O, which indicates the combination of silica (Si) and chitosan (N and O elements) on the surface of graphene oxide (C and O elements) in Figure 3c.

#### 3.1.3. SEM and Mapping Technique

Figure 3a illustrates the layers of GO in SEM images, and in comparison with Figure 3b, which shows the GOSCh, it can be understood that the surface of the GO has been occupied with chitosan and silica elements. It is difficult to show the silica on the SEM image, because of the small size of silica and the combination of materials on the surface of the graphene oxide sheets. For indicating the elements of the adsorbent, the mapping of SEM was used to shows in Figure 3c. Additionally, according to the mapping tests in Figure 3c, nitrogen and silicon dispersions are observed on the GO surface. The red and light purple plots show that the nitrogen and silica are uniformly distributed on the surface of the GO. These uniform distributions on the GO surface illustrates the successful interaction between silica and chitosan and the GO.

#### 3.1.4. Raman Spectroscopy Analysis

Figure 4 shows the GO and GOSCh Raman spectra. The GO oxidation rate can be exhibited through the differences in the D and G peak intensities in the Raman spectra. The D-band of the GO and GOSCh are determined at 1357.02 and 1353.06 cm^−1^, respectively. The D-band small shift to the left for the synthesised GOSCh (when compared to GO) might be attributed to the interaction between chitosan functional groups and the oxygen bonds of GO. The G-band peaks were at 1587.10 and 1588.09 cm^−1^ for GO, and the GOSCh can be described by the first order E^2^g phonon scattering from sp^2^ carbon particles [28]. As can be observed in Figure 4, the ID/IG degree for GO (0.855) is higher than for GOSCh (0.852). These effects suggest that further graphitic areas and more sp^2^ groups remain for GO in comparison to GOSCh. Furthermore, the GOSCh displayed peaks at 1488 cm^−1^ which can represent C–H vibrations [29] and at 2673.99 and 2914.9 cm^−1^, due to the presence of the chitosan band in the 2D area [30].

#### 3.1.5. Surface Area Analysis by Brunauer Emmett Teller (BET)

The surface areas for the GO and GOSCh are presented in Table 1. The surface area of GO was increased from 6.33 m^2^/g to 10.16 m^2^/g after mixing with silica and chitosan elements. The surface area size of GOSCh was increased as expected, because of the functionalising with organic groups of chitosan and silica. The greater specific surface area of GOSCh has a positive effect on bonding and chelating with heavy metal ions, so it makes the composite an attractive applicant for eliminating the Pb(II) and Cd(II) ions from aqueous solution. 

### 3.2. Batch Experiments of Adsorption

#### 3.2.1. Effect of Chitosan Percentage Ratio

In this study, the 20%, 40%, 60% and 80% ratio of chitosan were added to GO-SiO to find the optimum amount of chitosan for preparing GOSCh adsorbent. The adsorption of Pb(II) at natural pH and 60 mg/L initial concentration for Pb(II) was tested with different amounts of chitosan. Both of graphene oxide and chitosan have functional groups to interact with Pb(II). By mixing the chitosan with graphene oxide, some of these functional groups make bonds with each other. When the amount of chitosan is less, the adsorption is low, and when the amount of chitosan is high, the chitosan amine groups cover more functional groups of graphene oxide and decrease the adsorption ability. For this reason, finding the best amount of chitosan is necessary for mixing with graphene oxide. Table 2 shows the adsorption removal percentages of the adsorbent with different ratios of chitosan for removing Pb(II). This part of the study shows that the optimum amount of chitosan was obtained at 40% of chitosan. According to this study, the optimum ratio of chitosan was 40%, and in the higher ratio of chitosan, the adsorption removal percentage decreased, because of the occupied the free active sites and functional groups of GO with extra chitosan. 

#### 3.2.2. Effect of Adsorbent Dosage

Effect of graphene oxide-silica-chitosan (GOSCh) on Pb(II) adsorption was achieved in the range of 0.04, 0.06, 0.08, 0.1, 0.3, 0.5 and 0.9 g of GOSCh in the 50 mL initial concentrations. The influence of adsorbent dosage for removing the Pb(II) is illustrated in Figure 5. The outcomes showed that the adsorption capacities increased to the 0.08 g significantly and increased to 0.1 g slowly, and then stayed unchanged for Pb(II). Removing percentages were raised because of the expansion of the free sites and possible adsorbent surfaces that 0.08 g considers as optimum adsorbent dosages for Pb(II). By increasing the amount of adsorbent dosage, the amount of functional groups, binding site, surface area and interaction efficiency rise to a specific amount, and then increase slowly and remain constant. The adsorption percentage remained constant after 0.1 g is ascribed to the saturation of available adsorption functional groups in the adsorption procedure [31].

#### 3.2.3. Effect of Initial Concentration

In adsorption studies, the initial concentration is an essential factor which has a high influence on the adsorption capacity. In this investigation, aqueous solution with different concentrations of 10, 20, 40, 60, 80, 100 and 120 ppm were used in the adsorption process. To these solutions 0.08 g of GOSCh were added, in order to determine the optimum initial concentration. It can be observed that the adsorption capacity gradually increased to 40 ppm with 90% removal, and then decreased with increasing concentration, as shown in Figure 6. This suggests that the best adsorption capability of GOSCh is obtained with an initial Pb(II) concentration of 40 ppm in aqueous solution. The Pb(II) initial concentration was set at this value for the following experiments. The free binding sites of the adsorbent (GOSCh) are filled by suspended Pb(II) ions. As a consequence of this process, the adsorption capability of GOSCh reduced when initial concentration increased due to the fact that numerous adsorbent active sites were preoccupied with the Pb(II) ions [32]. The adsorbent of GOSCh has a specific number of functional groups and a certain Pi-Pi (п-п) performance. On the other hand, with increasing the initial concentrations, the amounts of Pb(II) ions are enhanced, and these Pb(II) ions occupy the functional groups of GOSCh like amine, amide, hydroxyl, carboxyl, ketone and epoxy. Then the removal ability of GOSCh adsorbent is higher in low value of concentration.

#### 3.2.4. Effect of pH 

Figure 7 presents the adsorption potentials for Pb(II) (0.08 g GOSCh and 40 ppm initial concentration) using GOSCh as an adsorbent at varying the pH from 3 to 8. It can be seen that the Pb(II) adsorption capacity increased from 87% at pH 3 to 92.5% at pH 6, and then decreased to 86.92% at pH 8. At a pH of 6 and higher, H^+^ and Pb(II) competition decrease with each other to occupy the free functional groups of the adsorbent [33]. Therefore, the remainder of the adsorption tests in this study were carried out at pH 6 with an initial Pb(II) concentration of 40 ppm. The adsorption phenomenon occurs due to electrostatic interactions between the adsorbent and the adsorbate [34]. On the other hand, the pH of adsorption solution has a direct effect on the ionisation and electrostatic interactions of the adsorbent functional groups. The functional groups of GOSCh, namely, amine, carboxyl, hydroxyl and epoxy have a direct effect on the Pb(II) ions adsorption, hence, the changes in pH of the aqueous solution is an essential behaviour factor in the adsorption process of the Pb(II) on the GOSCh. On the other hand, the π–π interaction of graphene oxide helps to increase the removal percentage with interacting between GOSCh and the Pb(II) ions of aqueous solution. Additionally, the solution includes the hydronium in the low pH, and this causes the competitive situation between H_3_O^+^ and Pb(II) ions to be adsorbed on the surface of adsorbent [35]. With increasing the pH, the removal capacity increased, because of the decreasing amount of hydronium, and there were most places on the adsorbent to adsorb the Pb(II) ions. The adsorption percentage of Pb(II) on adsorbent increased with increasing pH from 3.0 to 6.0, because of the decreasing the competition between Pb(II) and H_3_O^+^ for covering the functional groups in higher ph. In this range, the solution is mainly composed of Pb(II), and in the range of pH 7.5–9.0 it changes to Pb(OH)^+^ and at a higher pH, it becomes Pb(OH)_2_ and Pb(OH)_3_^−^. More negatively charged GO was detected at pH 3.0–6.0 and the removal ability of Pb(II) on the adsorbent increases because of the electrostatic attraction between the negative functional groups charge and the positive charge of Pb(II). The formation of precipitate of Pb(II) happen in high pH when it turns to (Pb(OH)_2_) [8].

#### 3.2.5. Effect of Contact Time

Contact time is a critical factor that influences the adsorption behaviour of the GOSCh and Pb(II). In this investigation, the effect of contact time on adsorption capacity was measured for an initial concentration of 40 ppm Pb(II) at a pH of 6 for various contact times of 5 to 140 min. Figure 8 illustrates the effect of contact time on adsorption. It can be observed that the adsorption capacity was significantly increased in the first 20 min and, subsequently, it elevated slowly for next 40 min and finally reached a steady state and a stable removal percentage at 80 min. 

This behaviour is due to the fact that a large number of functional groups of amine, amide, hydroxyl, carboxyl, ketone and epoxy of GOSCh are available in the form of active adsorption sites at the starting of the process [36]. However, as the time passes, the number of occupied active sites increases, hence, the adsorption rate reaches to a steady state stage, similar phenomenon was observed by a number of other authors [36,37,38]. The contact time has a significant effect on the enhancement of electrostatic and π–π interaction between the GOSCh and Pb(II) in aqueous solution [39]. Moreover, the constants amount of Pb(II) concentration in solution competitive to occupying the functional groups of chitosan and graphene oxide of GOSCh adsorbent. In the start of process, the amine, amide, hydroxyl, carboxyl, ketone and epoxy groups are free and accessible to interact with Pb(II) and, after occupying the surface functional groups, the Pb(II) ions attempt to penetrant to GOSCh, then the removal ability increased slowly and stabled.

### 3.3. Adsorption Isotherms

Isotherm models are analytical and mathematical patterns used to explain the diffusion and concentration of pollution within the solution and adsorbent phases. Isotherm models are based on some theories and assumptions that depend on the solid cover heterogeneity or homogeneity, surface character and the interaction feasibility between the pollution and adsorbent [40]. Additionally, adsorption isotherm models are important for demonstrating the adsorbent ability and efficiency. In this investigation, the Freundlich, Harkins-jura, Langmuir, Duldinin-Redushkevich and Temkin models [41] were applied to the experimental results; their parameters are shown in Table 3. The R^2^ were calculated and the error analysis used for analysing the linear regressive accuracy. So, the non-linear R^2^ and dynamic parameters of the various models have the least errors and for the linear formula is better to use the error analysis to obtain the better models [42].

One of the relative mathematical formulas of error analysis is a derivative of Marquardt’s percent standard deviation (MPSD) [43,44]:(3)$$MPSD=\overset{p}{\underset{t=1}{\sum }}{\left(\frac{X_{measure}-X_{calculated}}{X_{measure}}\right)}^{2}$$

Where, Q max (mg/g) is the adsorption capacity of each isotherm model, k is the Freundlich constant and n is the intensity constant. The constant of b (L/mg) in the Langmuir isotherm is the relationship between the adsorbate and the adsorbent binding sites. B, A and b are Harkins jura, D-R and Temkin constant. Regarding the constants displayed in Table 3, the best R^2^ values for the isotherm models considering adsorption of Pb(II) by GOSCh were obtained for the Temkin, Freundlich, and Langmuir models with R^2^ values of 0.94, 0.99 and 0.93, respectively. The Freundlich constant (n) is more than 1 and the Langmuir R_L_ is between 0 and 1 which indicates that this adsorption process happened in favourable form [30]. Additionally, the behaviour of adsorption is both mono layer and multi-layer. In this study, the maximum adsorption (Q max) of Pb(II) adsorption was 256.41(+/−4%) mg/g, calculated by the Langmuir model. This adsorption data and behaviour is related to the Pi-Pi (п-п) interaction and ion exchanging of functional groups of adsorbent and Pb(II). Figure 9 illustrates the isotherm models of the adsorption process.

### 3.4. Kinetics Study

Kinetic models are important for determining the mass transport and adsorption mechanism using the experimental results. The kinetic values and the related of pseudo-first-order, Pseudo-second-order, Elovich and intra-particle constant models were achieved by applying linear regression [12]. 

Where k, in the pseudo-first-order model, is the constant rate, k, in the pseudo-second-order model, is the kinetic rate constant and Q is the adsorption capacity of the adsorbent. The α and β parameters in the Elovich model show the adsorption rate and adsorption constant, respectively. Additionally, the k in the Intra-particle is the intra-particle’s diffusion rate constant.

According to Table 4 and Figure 10, it can be seen that the pseudo-second-order kinetic models fit the adsorption of Pb(II) by GOSCh with an R^2^ value of 0.999. The pseudo-second order kinetic model assumed that the process was chemisorption, meaning the adsorption occurred with the participation of chemisorptive bonds or electron exchange between Pb(II) and GOSCh.

### 3.5. Regeneration

Insufficient consideration has been given to the regeneration of used adsorbents after removing heavy metal ions. There are not many efficient studies on the regeneration of used adsorbents after heavy metal removal from a waste stream. The regeneration efficiency of the adsorbent is directly dependent on the ratios of available active sites on the structure of the adsorbent when the adsorption process is completed [45]. The regeneration of the adsorbent provides various benefits, like the reusability of the adsorbent, the recognition of the mechanism of adsorption, reduction of the secondary effluent and an increased economic process value [20]. In this study, the adsorbent was regenerated for five cycles using 0.1 M HCl, due to the protonation of the adsorbent surface by H_3_O^+^ in the acidic condition. The HCl could cause the removal of Pb(II), due to the positive charge on the adsorbent surface [46]. As can be seen in Figure 11, the removal percentage for GOSCh decreased from 96.25% to 87.75% after five cycles. 

In this investigation, adsorption of Pb(II) using synthesised adsorbent of GOSCh presented encouraging outcomes of the adsorbent with benchmarking against other studies that were conducted. The synthesised adsorbent of GOSCh displayed the highest potential of adsorption for removing Pb(II) with the Q max value of 256.41 (+/−4%) mg/g. The great adsorption capacity of the GOSCh in this investigation could be attributed to the various functional groups, the п-п interaction of GO and synergistic influences of chitosan, silica and GO particles, which raised the active sites of synthesised adsorbent. The maximum adsorption ability of Pb(II) is compared with other studies in Table 5, which illustrates the great powers of GOSCh adsorbent for removing the Pb(II). In other investigations, different materials were used like chitosan, activated carbon, graphene oxide with different methods, such as functionalisation, magnetisation, coating and making a composite of adsorbent. All of these methods and materials are suitable for removing Pb(II), because of their functional groups, ion exchanging and interacting with Pb(II) ions. While in this study, the chitosan and GO were used for their various functional groups for ion exchanging, п-п structure of GO and interactions with Pb(II). Furthermore, the silica was applied to increase the hydrophilicity of adsorbent with decreasing the resistance of the layer between aqueous solution and surface of adsorbent solid phase. These reasons help to increase the adsorption capacity and removal ability of GOSCh adsorbent in compare of other studies.

To benchmark the GOSCh adsorption capability of GO, GO-Si and GOCh were tested, and the results are shown in Figure 12. The obtained adsorption capacities of GO, GO-Si and GOCh were 8.12 mg/g, 12.5 mg/g and 24.06 mg/g (+/−2%) for Pb(II) respectively. The adsorption capability of the GO is related to the functional groups and п-п interaction of GO with Pb(II). After adding the silica, the silane groups help to increase the hydrophilicity of adsorbent and decrease the resistance of the exterior surface of adsorbent and bulk solution. Furthermore, chitosan with functional groups of amine and hydroxyl has been used in this investigation. These functional groups help to increase the adsorbent ability, and it can be seen that the adsorption capacities of GOSCh were enhanced by 33.7% (+/−0.2%) for Pb(II) removal.

## 4. Conclusions

In this study, the GOSCh adsorbent was synthesised successfully according to TEM, SEM, mapping, Raman, Zeta sizer and FT-IR analysis. FT-IR and Raman peaks illustrated the combination of silica and chitosan with GO, and the TEM, SEM and mapping analyses demonstrated a uniform dispersion of silica and chitosan on the GO layer. It could be understood that the functional groups of silane, amine, amide, hydroxyl, carboxyl, ketone and epoxy were set on the surface of graphene oxide sheets properly. The batch system of adsorption process was used to find the removal ability of GOSCh by different parameters of dosage, initial concentration, pH and contact time. The GOSCh proved to be effective in Pb(II) adsorption. Then, the isotherm and kinetic models were calculated to find the behaviour and mechanism adsorption of GOSCh adsorbent for removal of Pb(II). The maximum adsorption capacity of Pb(II) by GOSCh, calculated from a Langmuir model, was 256.41 (+/−4%) mg/g and was fit by a pseudo-second-order kinetic model. The π–π interaction and functional groups of the GO, chitosan, and silica are the main parameters affecting the high adsorption capability of the GOSCh. As the results of this interactions, the new synthesised material has a high potential for removing Pb(II) from aqueous solution.

## Figures and Tables

**Figure 1 polymers-12-01922-f001:**
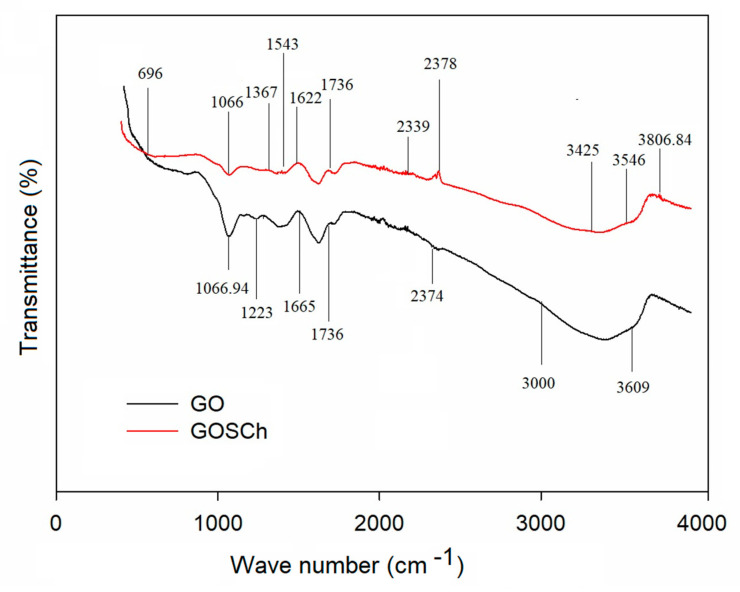
Graphene oxide (GO) and graphene-silica-chitosan (GOSCh) wavelength using Fourier-transform infrared spectroscopy (FT-IR) spectroscopy.

**Figure 2 polymers-12-01922-f002:**
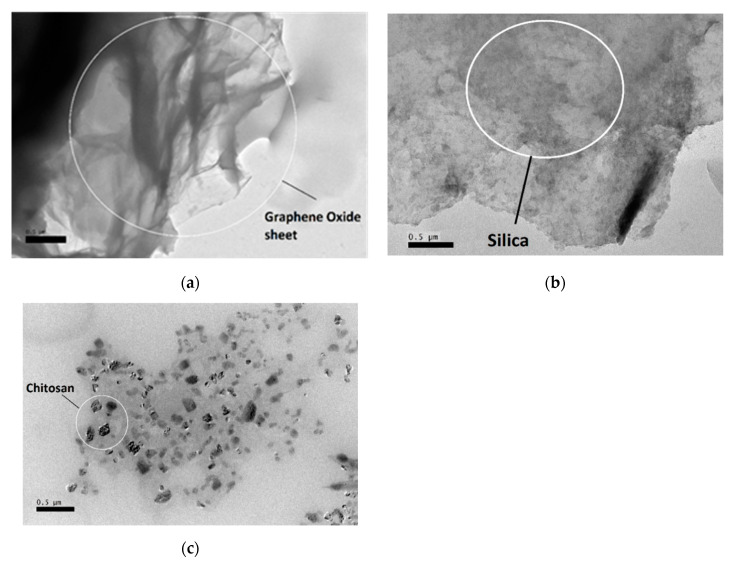
Transmission electron microscopy (TEM) analysis for (**a**) graphene oxide sheets; (**b**) uniform silica coating on the graphene oxide layer; (**c**) coated chitosan and silica on the graphene oxide sheets.

**Figure 3 polymers-12-01922-f003:**
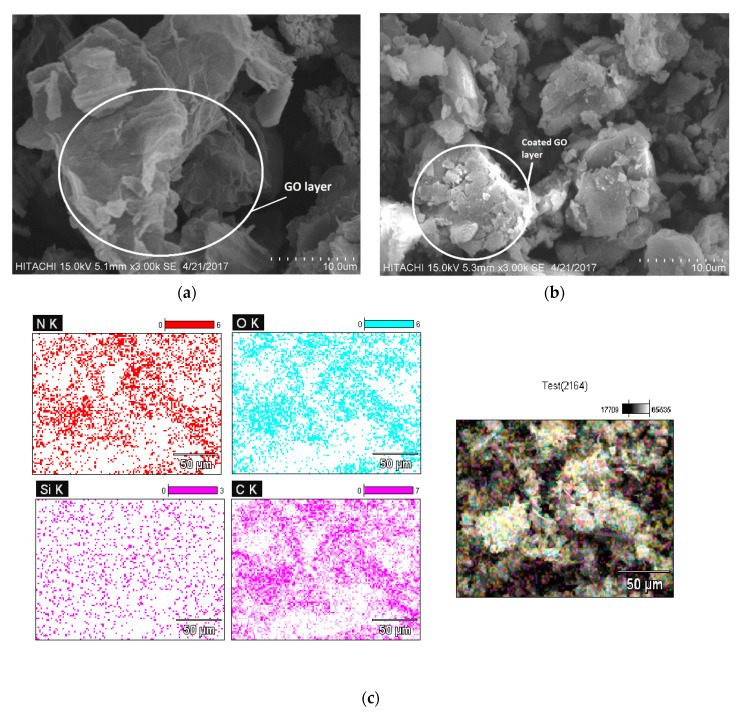
Scanning electron microscope (SEM) analysis of (**a**) graphene oxide sheets; (**b**) coated chitosan and silica on the graphene oxide sheets; (**c**) indicating the elements of the adsorbent by mapping of SEM.

**Figure 4 polymers-12-01922-f004:**
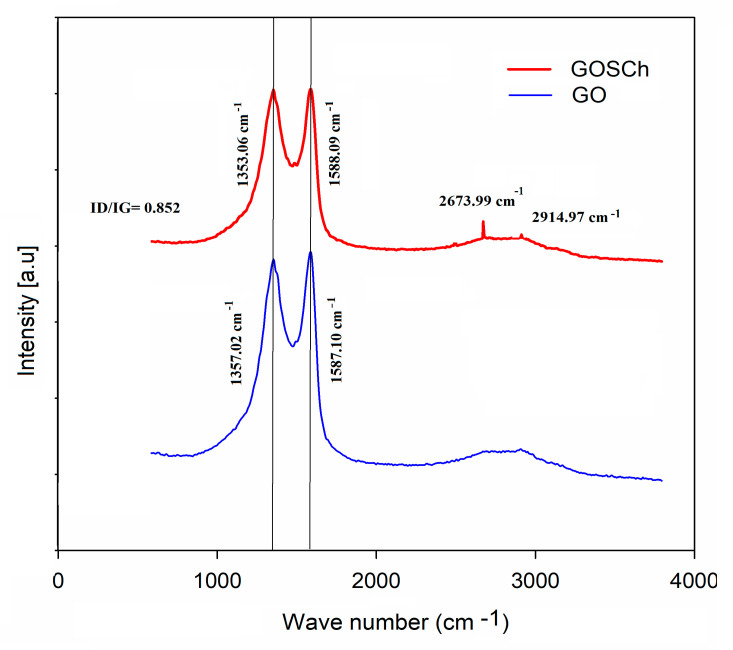
Raman spectroscopy analysis of GO and GOSCh.

**Figure 5 polymers-12-01922-f005:**
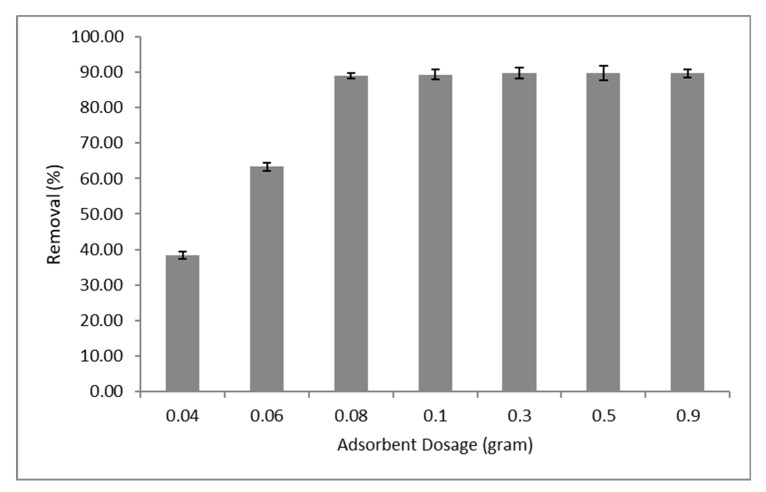
Adsorption capacities of Pb(II) by GOSCh with varied dosages from 0.04 to 0.9 grams.

**Figure 6 polymers-12-01922-f006:**
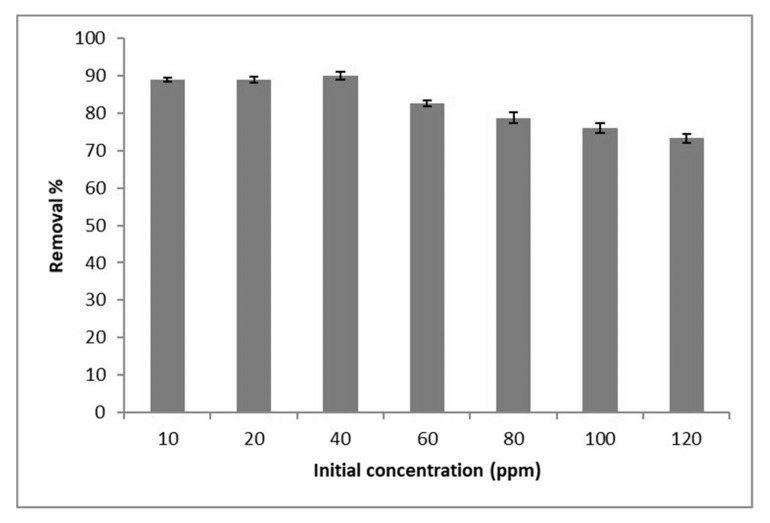
Adsorption capacities of Pb(II) by GOSCh with varies of concentrations from 10 ppm to 120 ppm.

**Figure 7 polymers-12-01922-f007:**
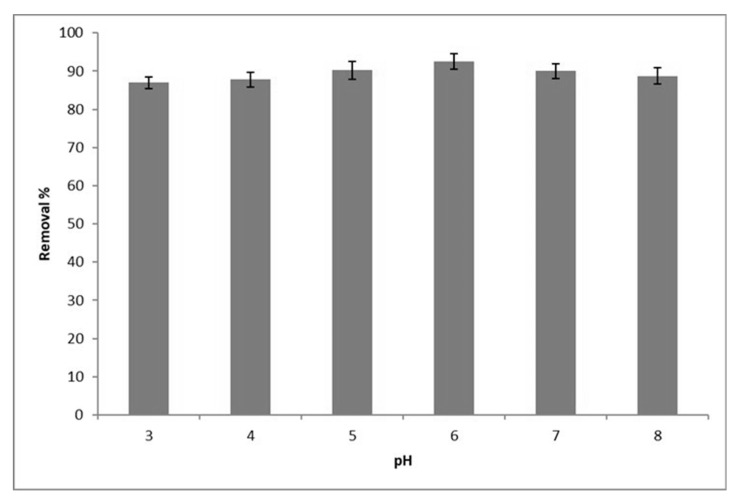
Adsorption capacities of Pb(II) by GOSCh at various pH values in the range of 3 to 8.

**Figure 8 polymers-12-01922-f008:**
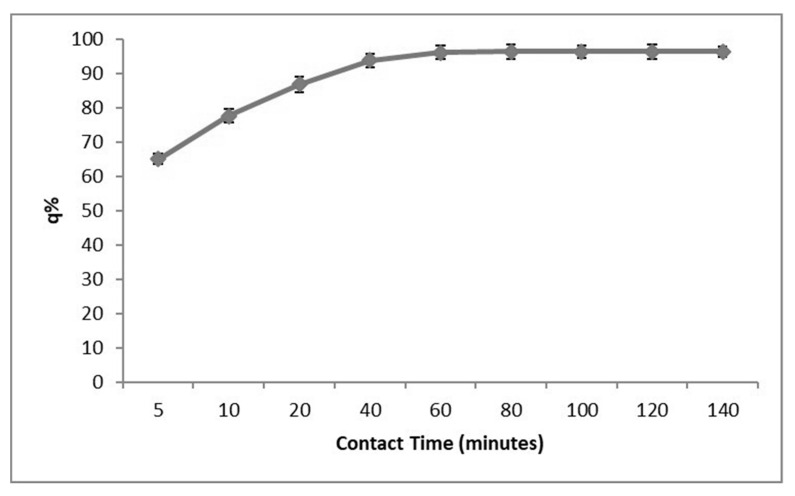
Adsorption capacities of Pb(II) by GOSCh at different times from 5 to 140 min.

**Figure 9 polymers-12-01922-f009:**
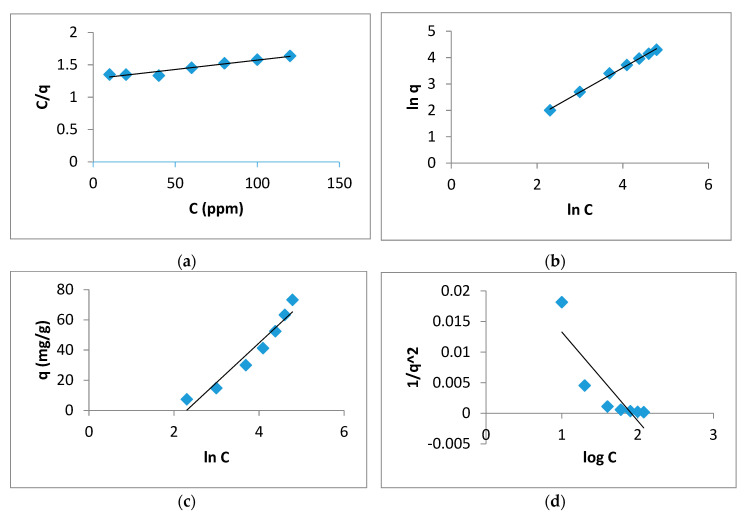
Linearised isotherm of (**a**) Langmuir; (**b**) Freundlich; (**c**) Temkin; (**d**) Harkins-Jura; (**e**) Duldinin-Redushkevich for the adsorption of Pb(II).

**Figure 10 polymers-12-01922-f010:**
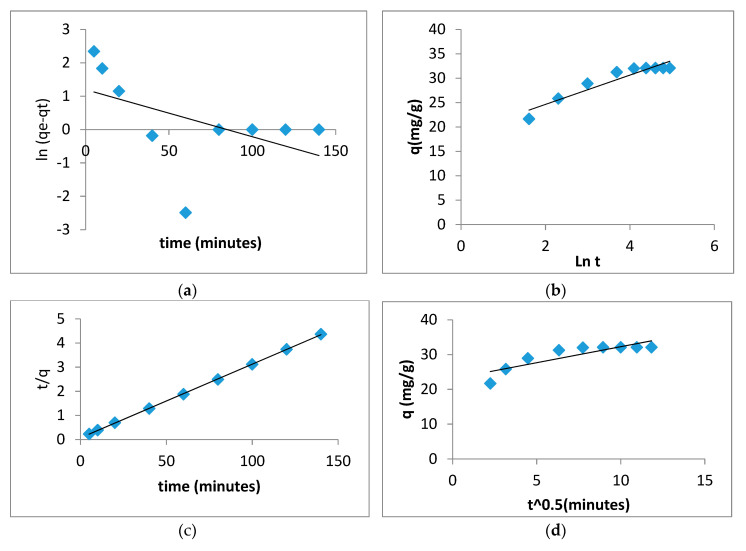
Kinetic models of (**a**) pseudo-first-order; (**b**) Elovich; (**c**) pseudo-first-order; (**d**) intra-particle adsorption for the adsorption of Pb(II) using GOSCh.

**Figure 11 polymers-12-01922-f011:**
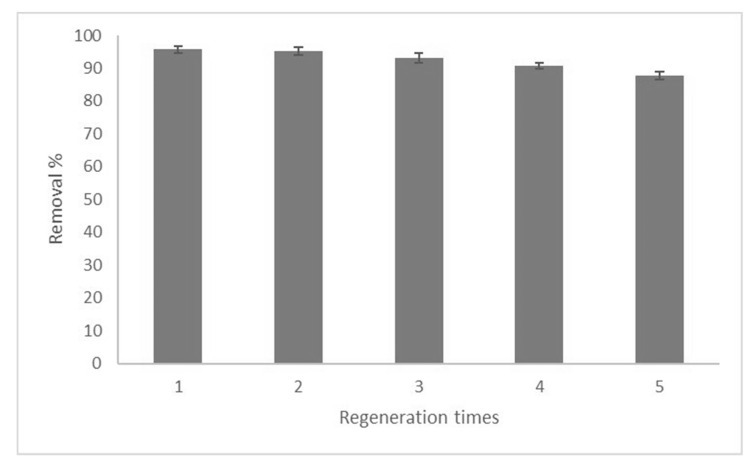
Removal percentage of Pb(II) by GOSCh after five cycles.

**Figure 12 polymers-12-01922-f012:**
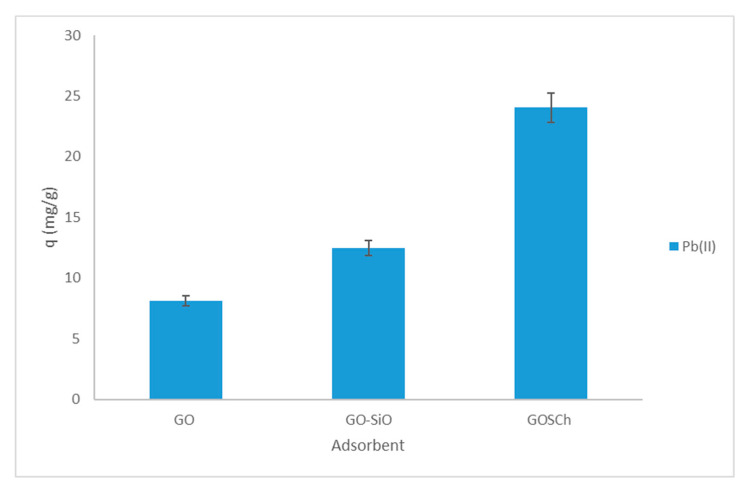
Comparison Adsorption Capacity of GOSCh, GO-SiO and GO.

**Table 1 polymers-12-01922-t001:** Structural parameters derived from nitrogen adsorption data for GO and GOSCh.

Adsorbent	SBET (m^2^/g)	Sext (m^2^/g)	Pore Volume (cm^3^/g)	Pore Size (nm) (Average Pore Diameter)
GO	6.33 (+/−2%)	6.044 (+/−2%)	1.27	61
GOSCh	10.16 (+/−2%)	10.16 (+/−2%)	1.078	70.8

**Table 2 polymers-12-01922-t002:** Percentage ratio of chitosan for coating on the graphene oxide-silica (GO-SiO).

1 g of GO-Si	Initial Concentration (mg/L)	Q% of Pb(II)	Q (mg/g) of Pb(II)
20% of chitosan	60	83.33 (+/−2%)	50 (+/−1)
40% of chitosan	60	88.34 (+/−2%)	53 (+/−1)
60% of chitosan	60	80.00 (+/−2%)	48 (+/−1)
80% of chitosan	60	75.00 (+/−2%)	45 (+/−1)

**Table 3 polymers-12-01922-t003:** Isotherm models parameters for adsorbing Pb(II) by GOSCh adsorbent.

Models	Parameters	Pb(II)
Freundlich	k (mg/g)	0.85
	1/n	0.92
	Q max (mg/g)	19.29
	R^2^	0.99
	MPSD	0.33
Langmuir	b (L/mg)	0.0022
	Rl	0.84
	Q max (mg/g)	256.41
	R^2^	0.93
	MPSD	0.63
Harkins-Jura	B	1.91
	A (L/g)	68.96
	R^2^	0.749
	MPSD	6.99
Duldinin-Redushkevich	B (mol^2^/kJ^2^)	0.00004
	Q max (mg/g)	47.322
	R^2^	0.796
	MPSD	2.77
Temkin	A (L/g)	0.647
	b	94.636
	R^2^	0.94
	MPSD	3.26

**Table 4 polymers-12-01922-t004:** Kinetic model parameters obtained for the adsorption of Pb(II) using GOSCh.

Models	Parameters	Pb(II)
Pseudo-first-order	K (h^−1^)	0.00046
	R^2^	0.5774
Pseudo-second-order	K (gmg^−1^ h^−1^)	0.0134
	Q (mg/g)	32.786
	R^2^	0.999
	MPSD	0.0033
Elovich	ἀ (mg/(g min))	161.65
	β (g/mg)	0.333
	R^2^	0.8958
	MPSD	7.52
Intra-particle	K (mgg^−1^ h^0.5^)	0.9215
	Ci	23.054
	R^2^	0.736
	MPSD	5.57

**Table 5 polymers-12-01922-t005:** Comparison between Pb(II) adsorption using GOSCh and other adsorbents.

Previous Study	Adsorbate	Modifying Method	Maximum Adsorption Capacity
Xu et al 2015 [7]	Pb(II)	Carboxyl-functionalised Chitosan Magnetic Microspheres	164.81 mg/g
Navid Saeidi et al 2015 [6]	Pb(II)	Graphene/Activated Carbon Composite	217.00 mg/g
Lee et al 2015 [47]	Pb(II)	Mn3O4-Coated Activated Carbon	59.52 mg/g
**This Study**	**Pb(II)**	**Graphene oxide-silica-chitosan**	**256.41 mg/g**

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
