# Peer review of "Synthesis and Characterisation of Graphene Oxide-Silica-Chitosan for Eliminating the Pb(II) from Aqueous Solution"

_polymers, 2020, doi:10.3390/polym12091922_

Round 1
Reviewer 1 Report
This manuscript provides a report on using synthesized Graphene Oxide-Silica-Chitosan for Pb (II) removal from aqueous solution. Freundlich, Temkin, and Langmuir isotherm models were fit to the results, and Langmuir isotherms model reached the maximum Pb (II) removal capacity. In general, the work is novel, and all required experiment is well done, the presentation of results and discussion are well organized. It seems to be suitable for publication. However, there are a few points that need to be clarified before it can be accepted.
- Introduction, state of the art of the technology needs to be better summarized, and the current status of the research gap and needs need to be clarified.
- The objective of the study needs to be strengthed, especially the novelty of the study.
- More discussion needs to be added to compare different researches.
- The quality of Fig 3 is too low.
Author Response
Please see the attachment. Also, all changes were highlighted in the manuscript to find easily. Thank you.
Response to Reviewer 1 Comments
Authors are thankful to the reviewers. Their comments will definitely help to improve the quality of the manuscript. Below are the answers to each specific issue raised up.
Reviewer 1
Point 1. Introduction, state of the art of the technology needs to be better summarized, and the current status of the research gap and needs need to be clarified.
Response1:
Thanks for this constructive suggestion. Finding the suitable materials with various functional groups and interaction ability is the important issue in preparing the adsorbent. Graphene oxide (GO) has attracted significant attention as a promising alternative adsorbent material due to its special properties, such as its large surface area(Liu et al., 2016), chemical stability and remarkable mechanical durability (Jiang et al., 2016). It has been shown to have a high adsorption potential for the elimination of Pb(II), Cd(II), and Co(II) from aqueous solutions (Saeidi et al., 2015). GO has chemically reactive oxygen functionality on the basal layers (hydroxyl and epoxy groups) and their edges (carboxylic acid) which increases reactions with other materials (Gao, 2015). And silica contains numerous silanol groups (Si-OH) on its outer shell that act as suitable nucleation and an anchor points for natural functionalization of silica.(Sun et al., 2008). And the surface of graphene oxide was coated by silica to improve the hydrophilicity features of composite (Kou & Gao, 2011). Another material that has been studied in this study was Chitosan with functional groups of -OH and -NH2 (Zhang et al., 2016). Chitosan can interact with heavy metal ions by chelation or ion exchange because of its functional groups (amino and hydroxyl groups)(Dubey et al., 2016).
Amendment: Lines 54-67
Point 2. The objective of the study needs to be strengthed, especially the novelty of the study.
Response2:
Thanks for this helpful suggestion. The suitable adsorbent needs the various amounts of functional groups to interact with heavy metal ions and make a proper situation to interact with heavy metal ions effortlessly. Hence in this study, for the novelty of this study, the silica was coated on the surface of GO and the combined with chitosan. Silica was used to increase the hydrophilicity of adsorbent with silane groups and let the adsorbent move inside the aqueous solution easily with reducing the resistance layer between the liquid solution and solid phase of adsorbent. And graphene oxide-silica and chitosan were combined to increasing the adsorbent efficiency by various functional groups of GO and chitosan, increasing the hydrophilicity by using silica, ion exchanging interaction between Pb(II) and functional groups of adsorbent and synergic effect of chitosan and graphene oxide interaction.
Amendment: Lines 72-80
Point 3. More discussion needs to be added to compare different researches.
Response3:
Thanks for this constructive comment. In other investigations, different materials were used like chitosan, activated carbon, graphene oxide with different methods such as functionalization, magnetisation, coating and make a composite of adsorbent. All of these methods and materials are suitable for removing Pb(II) because of their functional groups, ion exchanging and interacting with Pb(II) ions. While in this study, the chitosan and GO were used for their various functional groups for ion exchanging, п-п structure of GO and interactions with Pb(II). Furthermore, the silica was applied to increase the hydrophilicity of adsorbent with decreasing the resistance of the layer between aqueous solution and surface of adsorbent solid phase. These reasons help to increase the adsorption capacity and removal ability of GOSCh adsorbent in compare of other studies.
Amendment: Lines 366-375
Point 4. The quality of Fig 3 is too low.
Response4:
Thank you. Done. The original quality of Fig3 (C) is low. But it’s tried to increase the quality of Fig 3 image.
Amendment: Lines 176-177

Reviewer 2 Report
The manuscript entitled “Synthesis and Characterization of Graphene Oxide-Silica-Chitosan for Eliminating the Pb(II) from Aqueous Solution” by Azizkhani and coworkers describes the use of a graphene-based absorbant for the removal of Pb(II) ions. In particular, the graphene oxide was coated with silica in order to enhance its hydrophilicity, and was subsequently modified by various amounts of chitosan to promote the adsorption of Pb(II) ions. Based on Langmuir isotherms, the maximum capacity for Pb(II) removal was found to be 256.41 mg of Pb(II) per g of sorbent.
A variety of characterization techniques were employed, such as Fourier-transform infrared spectroscopy (FT-IR), Raman spectroscopy, scanning electron microscopy (SEM), transmission electron microscopy (TEM), as well as various mapping analysis methods. In addition, Langmuir, Temkin, and Freundlich isotherms were employed to fit the binding data and pseudo-second order kinetic models were also employed.
This manuscript would be of interest to researchers in polymer science, materials science, as well as supramolecular chemistry and also has environmental relevance. The research performed was generally systematic in nature and the results were generally well-supported by the findings provided by the authors.
There are some areas that could benefit from polishing, such as the grammar and writing as well as the presentation of numerical values (significant figures may need to be checked and error margins provided).
Overall, I believe that this manuscript is worthy of publication pending minor revisions. Some general suggestions are provided below.
Line 19: “various amount” can be changed to “various amounts”.
Line 27: The significant figures of the value 256.41 mg/g may need to be checked and possibly error margins should be provided.
Line 33: The phrase “and the rapid rate of urbanization Pb(II) to severe water contamination” is unclear.
Line 46: “0.01mg/L” can be changed to “0.01 mg/L”. Also a space should be inserted between the number and corresponding unit numerous other places in this manuscript.
Lines 65-66: The sentence “And with coating on the surface of graphene oxide Pb(II)s to improve the
hydrophilicity features of composite” is a little unclear.
Line 77: Text seems to be missing from the phrase “(Pi- Pi and Ion-exchange, and ....)”.
Line 83: The phrase “do ion exchanging” can possibly be changed to “undergo ion exchange”.
Line 96: “The 10 g of natural graphite added to the 5 g of sodium nitrate” can possibly be changed to “Natural graphite (10 g) was added to sodium nitrate (5 g)”.
Lines 97, 99 and 103: “2hr” can be changed to “2 h”.
Lines 98-99 review: The phrase “added to beaker and keep the temperature of the beaker under 10ºC with ice bath and
stirred for 2 hr” is unclear.
Line 111: “elements was detected” can be changed to “elements were detected”.
Line 125: “C0” should be changed to “C0” (with “0” written as a subscript).
Line 133: The significant figures of the values “1066.94 and 1223 cm” are not consistent with each other. Maybe this should be “1067 and 1223 cm”.
Line 150: “3425 cm-1may” should be changed to “3425 cm-1 may”.
Lines 151-153: “caused the enhancing of hydrophilicity and decrease the resistance between liquid solution and outside surface of solid adsorbent” can possibly be changed to “caused an enhancement of the hydrophilicity and decreased the resistance between the liquid solution and the outside surface of solid adsorbent”.
Line 162: “dramatic and feasible to see” can possibly be changed to “dramatic and clearly visible”.
Line 163: “same results were” can possibly be changed to “similar results” or “the same results”.
Lines 171-174: The sentence appearing in these lines is shown in a larger text font than the other text in this manuscript.
Lines 172-174: The sentence “And for indicating the elements of the adsorbent, it used the mapping of SEM that shows in the part of (c) in the Figure 3.” is unclear.
Lines 201-202, Table 1: Error margins may be needed for the numerical values presented in Table 1.
Line 206: “was tested at different amount of” can be changed to “was tested with different amounts of”.
Line 214: “According to this study, indicate that the optimum ratio of” can possibly be changed to “According to this study, the optimum ratio of” or “This study indicates that the optimum ratio of”.
Lines 217-218, Table 2: Error margins may be needed for the numerical values. Also the significant figures of the values reported in the “Q%” column seem to be inconsistent as some numbers are reported with 2 decimal places while others are reported as whole numbers.
Lines 219-220: The phrase “adsorption was achieved at 0.04, 0.06, 0.08, 0.1, 0.3, 0.5 and 0.9 gram in the 50 ml initial concentrations.” is unclear.
Line 226: “efficiency rise till specific amount and then increase slowly and stayed constant.” can possible be changed to “efficiency rise to a specific amount and then increase slowly and remain constant.” or “efficiency rise until a specific value is reached and then increase slowly and remain constant.”.
Lines 241-242: “that a numerous adsorbent active sites are” can be changed to “that numerous adsorbent active sites were”.
Lines 242-246: The last two sentences of this paragraph are a little unclear and difficult to follow.
Line 265, 269: “H3O+” should be changed to “H3O+” (with “3” as a superscript and “+” as a superscript. The authors will need to check the chemical formula names throughout the manuscript as there are other places where the chemical formulas are written incorrectly, such as “Pb(OH)2” and Pb(OH)3-” in lines 271 and 274.
Line 270: “solution is main” can possibly be changed to “solution is mainly composed of”.
Line 271: “and in higher ph turn to” can be changed to “and at a higher pH becomes”.
Line 276, Figure 7 caption: “in varies pH value” can be changed to “at various pH values”.
Line 281: “adsorption. it can be” can be changed to “adsorption. It can be”.
Line 284: “this behaviour is” should be changed to “This behaviour is”.
Line 295: The phrase “sharpness decreased and stabled” is unclear.
Line 308: “of the various model” can possibly be changed to “of the various models”.
Line 310: “One of the relative mathematical formula” can be changed to “One of the relative mathematical formulas”.
Line 315: “in Langmuir is the” can be changed to “in the Langmuir isotherm is the”.
Line 320: “which indicate that” can possibly be changed to “which indicates that”.
Line 322: The significant figures for the value “256.41 mg/g,” may need to be checked and error margins may be needed.
Line 324: “Figures 9 illustrate” can be changed to “Figure 9 illustrates”.
Lines 330-331: may need references for the kinetic models mentioned here.
Line 334: “Pseudo-first-order is the constant rate, k in the Pseudo-second-order is” can be changed to “pseudo-first-order model is the constant rate, k in the pseudo-second-order model is”.
Lines 335-336: “The α and β in the Elovich shows the adsorption rate and adsorption constant respectively,” should be changed to “The α and β parametsrs in the Elovich model show the adsorption rate and adsorption constant, respectively.” (Also a period instead of a comma is needed at the end of this sentence.
Line 336: The phrase “Also the in the Intra-particle is the intra-particle” is unclear. Text seems to be missing.
Line 337: “diffusion rat constant” should be changed to “diffusion rate constant”.
Line 352: “0.1M HCl” should be “0.1 M HCl”.
Line 361: The significant figures for the value 256.41 mg/g may need to be checked and possibly error margins may be needed.
Line 362: “could be described by the” can possibly be changed to “could be attributed to the”.
Lines 366-367, Table 5: Error margins may be needed for the adsorption capacity values.
Lines 368-369: The significant figures for the adsorption capacities need to be checked (they seem to be inconsistent with one another) and error margins may be needed.
Line 374: “of GOSCh enhanced 33.7%” can possibly be changed to “of GOSCh were enhanced by 33.7%” (Also the significant figures of 33.7% may need to be checked and error margins may be needed).
Lines 387-389: The significant figures of the value “256.41 mg/g” may need to be checked and error margins possibly provided.
Author Response
Please see the attachment. All changes were highlighted in the manuscript to find easily.
Response to Reviewer 2 Comments
Authors are thankful to the reviewers. Their comments will definitely help to improve the quality of the manuscript. Below are the answers to each specific issue raised up.
Point 1. English and editing corrections.
Response 1:
Thank you so much for these helpful comments, the comments about the English and editing were checked carefully and have been changed in the whole of article.
Amendment: Lines: 19/ 33/ 46/ 71/ 82/ 95-98/ 102/ 110/ 124/ 150-152/ 161/ 170-173/ 205/ 213/ 218/ 225/ 242-246/ 254/265/269-271/ 277/282/285/ 295-296/ 309/ 311/ 316/321/ 326/ 335-338/ 353/ 363.
Point 2. The significant figures of the value 256.41 mg/g may need to be checked and possibly error margins should be provided.
Response 2:
Thanks for this constructive suggestion. This value is maximum adsorption capacity which calculated from Langmuir isotherm model with formula of “Ce/qe = 1/(KLqm) + Ce/qm “. Standard error margin is about a +/- 4%. (256.41 +/- 4%)
Amendment: line 27/ 324/ 362/ 398.
Point 3. Table 1: Error margins may be needed for the numerical values presented in Table 1.
Response 3:
Thanks for this suggestion. These values were achieved from micrometritics analyzer instrument without mentioning the margins error but the error is in limited +/- 2% regards the standard sample along with instrument.
Amendment: Lines 200-201
Point 4. may need references for the kinetic models mentioned here.
Response 4: done. Ref: (Zewail & Yousef, 2015)
Zewail, T. M., & Yousef, N. S. (2015). Kinetic study of heavy metal ions removal by ion exchange in batch conical air spouted bed. Alexandria Engineering Journal, 54(1), 83–90. https://doi.org/10.1016/j.aej.2014.11.008
Amendment: Lines 333
Point 5. Table 2: Error margins may be needed for the numerical values.
Response 5:
Thanks for this suggestion. The error margins were added.
Amendment: Lines 216.
Point 6. Table 5: Error margins may be needed for the adsorption capacity values.
Response 6: Thank you for comment, Table 5 is the comparison of our study with others and they don’t have any error margin values in their study and just mentioned the fix number.
Amendment: Lines 376.
Point 7. Lines 387-389: The significant figures of the value “256.41 mg/g” may need to be checked and error margins possibly provided.
Response 7: Thank you for comment, this part is related to calculating and comparing the adsorption capacities of GO, GO-Si, and GOCh materials with formula of , and the 256.41 mg/g is related to maximum adsorption capacity which achieved by Langmuir isotherm model. Langmuir isotherm model shows the monolayer behaviour and maximum adsorption ability that be continued in long term and researchers use this model for Maximum adsorption capacity (Qmax) and use the for adsorption capacity.
Amendment: Lines 387-389.

Round 2
Reviewer 1 Report
The revised version improved a lot and all my comments are well stressed.
I have no further comments.
This manuscript is a resubmission of an earlier submission. The following is a list of the peer review reports and author responses from that submission.
Round 1
Reviewer 1 Report
This paper (polymers-833687) deals with developing graphene oxide-silica-chitosan composites for removal of Pb from aqueous solution. In my opinion, this paper can be published in Polymers after major revision.
Specific comments:
- The major concern is that the authors failed to explain the rationality of the design of graphene oxide-silica-chitosan composites. Both graphene oxide and chitosan can adsorb heavy metal ions, but why is the combination of graphene oxide and chitosan necessary? The Introduction looks like a mixture of different information. There is a lack of logical linkage between each component. The authors should rewrite the Introduction.
- How did the authors collect graphene oxide-silica-chitosan composites after adsorption? More details about Section 2.3 Adsorption Process are needed.
- The authors didn’t mention anything about the characterization of samples. They should add them in the manuscript.
- For 3.1.2, without the description of TEM test, it is difficult to judge if the images are silica and chitosan.
- For 3.1.3, the coated GO layer is not clear. The distribution of Si is different from the other elements. It is difficult to judge the distribution of Si. The authors should provide a better evidence to support their conclusion.
- According to SEM and TEM images, the size of composites is obviously larger than 1000 nm. Zeta Nano sizer is not a suitable equipment to measure such a large particle, and the results in Figure 5 are meaningless.
- Statistical analysis is needed to indicate the significant difference of the results.
Author Response
Response to Reviewer 1 Comments
The authors are thankful to the reviewers. Their comments will definitely help to improve the quality of the manuscript. Below are the answers to each specific issue raised up.
Reviewer 1
Point 1. The major concern is that the authors failed to explain the rationality of the design of graphene oxide-silica-chitosan composites. Both graphene oxide and chitosan can adsorb heavy metal ions, but why is the combination of graphene oxide and chitosan necessary? The Introduction looks like a mixture of different information. There is a lack of logical linkage between each component. The authors should rewrite the Introduction.
Response1:
Thank you so much for this constructive comment, although Both of the graphene oxide and chitosan are adsorbent with their specific functional groups to remove heavy metal ions. There is a lack of information in the literature about the combined capability of these adsorbents to remove heavy metal from aqueous solution. Our hypothesis was that decoration of chitosan and silica will enhance the adsorption capability, by adding different mechanisms of adsorption (Pi- Pi and Ion-exchange, and ....) and enhancing the adsorption capacity and resignation capability. Hence in this study, graphene oxide-silica and chitosan were combined to increasing the adsorbent efficiency by various functional groups, increasing the hydrophilicity by using silica, ion exchanging interaction between Pb(II) and functional groups of adsorbent and synergic effect of chitosan and graphene oxide interaction. The results show significant enhancement toward the removal of Pb(II).
Some parts of the introduction were changed to better explain the problem statement and goals of this paper
Amendment: Page 2, line 76-82
Point 2. How did the authors collect graphene oxide-silica-chitosan composites after adsorption? More details about Section 2.3 Adsorption Process are needed.
Response2:
Thanks for highlighting this issue. After adsorbing the Pb(II) in the solution by GOSCh adsorbent, the samples were centrifuged and passed from the filter to separate the occupied adsorbent from the final solution. The concentration of solution after adsorption was decreased and set on the 5ppm concentration which is the maximum standard of Atomic Absorption Spectroscopy for eliminating the heavy metal ions.
Amendment: Page 3, line 119-121
Point 3. The authors didn’t mention anything about the characterization of samples. They should add them in the manuscript.
Response3:
Thanks for this constructive suggestion. The part of 3.1 in the results and discussion segment is related to characterization of GOSCh sample using FTIR, TEM, SEM, RAMAN and Zeta sizer. The functional groups and elements was detected by FTIR, the surface analysis, morphology and elements was studied using SEM, mapping, TEM, and RAMAN.
Amendment: Page 3, line 109-112 and PART 3.1
Point 4. For 3.1.2, without the description of TEM test, it is difficult to judge if the images are silica and chitosan.
Response4:
Thanks for this constructive suggestion. The TEM test shows the surface changing not the element. In this regard step by step, TEM analysis was provided, the changes are made on the surface of graphene oxide sheets are dramatic and feasible to see. Additionally, same results were supported using the SEM mapping, which shows the presence of elements such as Si, N, C, and O which indicates the combination of Silica (Si) and Chitosan (N and O elements) on the surface of Graphene oxide (C and O elements) in the part of c in Figure 3. For your more information, some papers with similar TEM results were referenced here: (Xue, Xu, Baig, & Xu, 2016), (Liu et al., 2015), (Kou & Gao, 2011), (Kumar & Jiang, 2016). Furthermore, some additional explanation was added to the paper.
Amendment: Page 4, line 159-163
Point 5. For 3.1.3, the coated GO layer is not clear. The distribution of Si is different from the other elements. It is difficult to judge the distribution of Si. The authors should provide a better evidence to support their conclusion.
Response5:
Thanks for highlighting this issue. It is true that, it’s difficult to show the silica on the SEM image because of small size of silica and combination of materials on the surface of the Graphene oxide sheets. And we don’t want to show the elements on the SEM image in parts of (a) and (b), we are showing the changes of the surface. But for indicating the elements of the adsorbent, it used the mapping of SEM that shows in the part of (c) in the Figure 3. For your more information, some papers with SEM were referenced here: (Chen, Chen, Bai, & Li, 2013), (Kumar & Jiang, 2016), (Zhang, Luo, Liu, Fang, & Geng, 2016). Furthermore, some additional explanation was added to the paper.
Amendment: Page 5, line 169-170
Point 6. According to SEM and TEM images, the size of composites is obviously larger than 1000 nm. Zeta Nano sizer is not a suitable equipment to measure such a large particle, and the results in Figure 5 are meaningless.
Response6:
Thanks for the suggestions. We have amended our statement throughout the manuscript to avoid quick conclusions to be made.
We have removed the sentences and part of 3.1.5 as follow from the manuscript:
3.1.5. Zeta Sizer Analysis of Graphene Oxide and GOSCh
Regards the Fig. 5 and Table 1 display the particle size analyzer and polydispersity (PdI) of GO and GOSCh. The particle size distribution was described as PdI, which in this study are in the range of 0–1 and shows the homogeneity of distribution. Values Close to 1 show heterogeneity and less than 0.5 displayed more homogeneity (Naggar et al., 2015). The particle size value of GO and GOSCh are 547 and 837.1 nm with PdI values of 0.63 and 0.22. The particle size and polydispersity characteristic of GOSCh obtain a value at 837.1 nm as well as lower PdI (0.22) indicating good uniformity of GOSCh solution and more homogeneity of the formed nanoparticulate system (Hebeish et. al., 2014). The particle size of GOSCh is larger than GO,because of the combining and coating the silica and chitosan on the surface of graphene oxide.
Figure 5. Size distribution by intensity of (a) GO and (b) GOSCh using
Zeta Nano sizer.
Table 1. Particle size and PdI of the adsorbent.
|
Material |
Particle size (nm) |
PdI |
|
GO |
547 |
0.63 |
|
GOSCh |
837.1 |
0.22 |
Point 7. Statistical analysis is needed to indicate the significant difference of the results.
Response7:
Thank you for your thoughtful comment. We agree with reviewer hence the statistical step by step of adsorption is applied to analyse the effects of different amount of chitosan, initial concentration of Pb(II), ph, and contact time were tested and the results were provided. a High correlation coefficient R2 value shows the satisfaction of process which calculated from isotherm and kinetic models. The R2 were calculated and the error analysis used for analyzing the linear regressive accuracy. So the non-linear R2 and dynamic parameters of the various model have the least errors and for the linear formula is better to use the error analysis to obtain the better models (Abdulla, 2009).
One of the relative mathematical formula of error analysis is a derivative of Marquardt’s Percent Standard Deviation (MPSD) (Mahmoud et.al., 2017; Xu rt. al., 2013):
Isotherm models statistical for adsorbing Pb(II) by GOSCh adsorbent.
|
Models |
Parameters |
Pb(II) |
|
Freundlich |
R2 |
0.99 |
|
MPSD |
0.33 |
|
|
Langmuir |
R2 |
0.93 |
|
MPSD |
0.63 |
|
|
Harkins-Jura |
R2 |
0.75 |
|
MPSD |
6.99 |
|
|
Duldinin-Redushkevich |
R2 |
0.79 |
|
MPSD |
2.77 |
|
|
Temkin |
R2 |
0.94 |
|
MPSD |
3.26 |
Kinetic model statistical obtained for the adsorption of Pb(II) using GOSCh.
|
Models |
Parameters |
Pb(II) |
|
Pseudo-first-order
|
R2 |
0.17 |
|
Pseudo-second-order
|
R2 |
0.99 |
|
MPSD |
0.0033 |
|
|
Elovich |
R2 |
0.89 |
|
MPSD |
7.52 |
|
|
Intra-particle |
R2 |
0.73 |
|
MPSD |
5.57 |
Amendment: Page 10, line 296-302 and Table 2; Page 12, Table 3
References:
Abdulla, F. a. (2009). Roof rainwater harvesting systems for household water supply in Jordan. Desalination, 243(September 2015), 195–207. https://doi.org/10.1016/j.desal.200
Chen, Y., Chen, L., Bai, H., & Li, L. (2013). Graphene oxide–chitosan composite hydrogels as broad-spectrum adsorbents for water purification. Journal of Materials Chemistry A, 1(6), 1992. https://doi.org/10.1039/c2ta00406b
Kou, L., & Gao, C. (2011). Making silica nanoparticle-covered graphene oxide nanohybrids as general building blocks for large-area superhydrophilic coatings. Nanoscale, 3(2), 519–528. https://doi.org/10.1039/c0nr00609b
Kumar, A. S. K., & Jiang, S. J. (2016). Chitosan-functionalized graphene oxide: A novel adsorbent an efficient adsorption of arsenic from aqueous solution. Journal of Environmental Chemical Engineering, 4(2), 1698–1713. https://doi.org/10.1016/j.jece.2016.02.035
Liu, Y., Li, W., Shen, D., Wang, C., Li, X., Pal, M., … Zhao, D. (2015). Synthesis of Mesoporous Silica / Reduced Graphene Oxide Sandwich – Like Sheets with Enlarged and “ Funneling ” Mesochannels, 1–12. https://doi.org/10.1021/acs.chemmater.5b01812
Mahmoud, A. S., Saryel-deen, R. A., Mostafa, M. K., & Peters, R. W. (2017). Artificial Intelligence for Organochlorine Pesticides Removal from Aqueous Solutions using Entrapped nZVI in Alginate Biopolymer Artificial Intelligence for Organochlorine Pesticides Removal from Aqueous Solutions using Entrapped nZVI in Alginate Biopolym, (January 2018).
Xu, Z., Cai, J., & Pan, B. (2013). Mathematically modeling fixed-bed adsorption in aqueous systems. Journal of Zhejiang University SCIENCE A, 14(3), 155–176. https://doi.org/10.1631/jzus.A1300029
Xue, X., Xu, J., Baig, S. A., & Xu, X. (2016). Synthesis of graphene oxide nanosheets for the removal of Cd(II) ions from acidic aqueous solutions. Journal of the Taiwan Institute of Chemical Engineers, 59, 365–372. https://doi.org/10.1016/j.jtice.2015.08.019
Zhang, L., Luo, H., Liu, P., Fang, W., & Geng, J. (2016). A novel modified graphene oxide/chitosan composite used as an adsorbent for Cr (VI) in aqueous solutions. International Journal of Biological Macromolecules, (Vi). https://doi.org/10.1016/j.ijbiomac.2016.03.027

Reviewer 2 Report
Overall, the paper is not a novel to be published in this prestigious journal
1. The terminology should be unified such as lead(II) and Pb(II).
2. Authors must explain the reason for the best performance in 40% of chitosan.
3. It’s hard to agree with pH experiments data. Pb(II) would precipitate above pH 6.
4. The discussions are seriously lacking.
Author Response
Response to Reviewer 2 Comments
Authors are thankful to the reviewers. Their comments will definitely help to improve the quality of the manuscript. Below are the answers to each specific issue raised up.
Point. Overall, the paper is not a novel to be published in this prestigious journal
Response1:
Thank you for your thoughtful comment. We agree with reviewer that the adsorbent of graphene oxide-chitosan was prepared and published in various journals but in this study, the silica was coated on the graphene oxide surface to increase the hydrophilicity and conquers to resistance part between outside of adsorbent and solution. The silica interacts with solution because of the high hydrophilicity and let the functional groups of graphene oxide and chitosan work perfectly then interact and ion exchanging between adsorbent and Pb(II) done easily and enhance the adsorption capacity. As shown in figure one the adsorption capacity of this material is superior to that of GO and Chitosan hence we truly believe this study can improve the body of knowledge, and help of other researchers applying this material to other areas such as dye adsorption and extra.
Amendment: Page 14, line 363-372 and Figure 12
Point 1. The terminology should be unified such as lead(II) and Pb(II).
Response1:
Thanks for highlighting this issue. Authors would like to apologizes for this mistake all Lead(II) were replaced by Pb(II).
Point 2. Authors must explain the reason for the best performance in 40% of chitosan.
Response2:
Thanks for this constructive suggestion. Both of Graphene oxide and Chitosan have functional groups to interact with Pb(II). And with mixing the chitosan with graphene oxide, some of these functional groups like namely amine groups of chitosan and hydroxyl and carboxyl groups of GO make bond to each other. the authors actually tested many different ratios of chitosan to graphene oxide and the results showed when the amount of chitosan is less the adsorption is low and when the amount of Chitosan is high, the Chitosan cover more functional groups of Graphene oxide and decrease the adsorption ability. For this reason, finding the best amount of Chitosan is necessary to mixing with Graphene oxide.
Amendment: Page 7, line 194-200
Point 3. It’s hard to agree with pH experiments data. Pb(II) would precipitate above pH 6.
Response3:
Thanks for the guidance and idea given. The adsorption percentage of Pb(II) on adsorbent increased with increasing pH from 3.0 to 6.0 because of the decreasing the competition between Pb(II) and H3O+ for covering the functional groups in higher ph. In this range, the solution is main Pb(II) and in the range at ph 7.5–9.0 it’s changed to Pb(OH)+ and in higher ph turn to Pb(OH)2 and Pb(OH)3−. And more negatively charged of GO was detected at pH 3.0-6.0 and removal ability of Pb(II) on adsorbent increase because of the electrostatic attraction between negative functional groups charge and positive charge of Pb(II). The formation of precipitate of Pb(II) happen in high ph when it turns to (Pb(OH)2) (Huang & Pan, 2016).
Amendment: Page 9, line 256-264
Point 4. The discussions are seriously lacking.
Response4:
Thanks for the kind guidance. We have added some relevant discussions in the different sections of the revised manuscript.
Amendment: Page 4, line 149-152- Page 4, line 159-163- Page 5, line 169-170- Page 7, line 194-200- Page 8, line 230-234- Page 9, line 274-275 and 280-285- Page 10, line 296-302 and Table 2 and Table 3 added the MPSD- Page 11, line 315-316- Page 14, line 378-383.
Reference:
Huang, X., & Pan, M. (2016). The highly efficient adsorption of Pb(II) on graphene oxides: A process combined by batch experiments and modeling techniques. Journal of Molecular Liquids, 215, 410–416. https://doi.org/10.1016/j.molliq.2015.12.061

Round 2
Reviewer 1 Report
The authors didn’t well respond to comments from the previous review. Therefore, in my opinion, this paper cannot be published in Polymers.
Point 2. How did the authors collect graphene oxide-silica-chitosan composites after adsorption? More details about Section 2.3 Adsorption Process are needed.
Response2:
Thanks for highlighting this issue. After adsorbing the Pb(II) in the solution by GOSCh adsorbent, the samples were centrifuged and passed from the filter to separate the occupied adsorbent from the final solution. The concentration of solution after adsorption was decreased and set on the 5ppm concentration which is the maximum standard of Atomic Absorption Spectroscopy for eliminating the heavy metal ions.
Amendment: Page 3, line 119-121
For Point 2, the speed and time of centrifugation should be indicated.
Point 3. The authors didn’t mention anything about the characterization of samples. They should add them in the manuscript.
Response3:
Thanks for this constructive suggestion. The part of 3.1 in the results and discussion segment is related to characterization of GOSCh sample using FTIR, TEM, SEM, RAMAN and Zeta sizer. The functional groups and elements was detected by FTIR, the surface analysis, morphology and elements was studied using SEM, mapping, TEM and RAMAN.
Amendment: Page 3, line 109-112 and PART 3.1
For Point 3, it is not enough to only mention the names of tests. Detailed descriptions about how the authors did these tests are needed.
Point 4. For 3.1.2, without the description of TEM test, it is difficult to judge if the images are silica and chitosan.
Response4:
Thanks for this constructive suggestion. The TEM test shows the surface changing not the element. In this regard step by step, TEM analysis was provided, the changes are made on the surface of graphene oxide sheets are dramatic and feasible to see. Additionally, same results were supported using the SEM mapping, which shows the presence of elements such as Si, N, C, and O which indicates the combination of Silica (Si) and Chitosan (N and O elements) on the surface of Graphene oxide (C and O elements) in the part of c in Figure 3. For your more information, some papers with similar TEM results were referenced here: (Xue, Xu, Baig, & Xu, 2016), (Liu et al., 2015), (Kou & Gao, 2011), (Kumar & Jiang, 2016). Furthermore, some additional explanation was added to the paper.
Amendment: Page 4, line 159-163
For Point 4, the authors still did not mention how they prepared the TEM samples and did the test.
Point 5. For 3.1.3, the coated GO layer is not clear. The distribution of Si is different from the other elements. It is difficult to judge the distribution of Si. The authors should provide a better evidence to support their conclusion.
Response5:
Thanks for highlighting this issue. It is true that, it’s difficult to show the silica on the SEM image because of small size of silica and combination of materials on the surface of the Graphene oxide sheets. And we don’t want to show the elements on the SEM image in parts of (a) and (b), we are showing the changes of the surface. But for indicating the elements of the adsorbent, it used the mapping of SEM that shows in the part of (c) in the Figure 3. For your more information, some papers with SEM were referenced here: (Chen, Chen, Bai, & Li, 2013), (Kumar & Jiang, 2016), (Zhang, Luo, Liu, Fang, & Geng, 2016). Furthermore, some additional explanation was added to the paper.
Amendment: Page 5, line 169-170
For Point 5, no better evidence is provided to support the conclusion.
Point 7. Statistical analysis is needed to indicate the significant difference of the results.
Response7:
Thank you for your thoughtful comment. We agree with reviewer hence the statistical step by step of adsorption is applied to analyse the effects of different amount of chitosan, initial concentration of Pb(II), ph, and contact time were tested and the results were provided. a High correlation coefficient R2 value shows the satisfaction of process which calculated from isotherm and kinetic models. The R2 were calculated and the error analysis used for analyzing the linear regressive accuracy. So the non-linear R2 and dynamic parameters of the various model have the least errors and for the linear formula is better to use the error analysis to obtain the better models (Abdulla, 2009).
One of the relative mathematical formula of error analysis is a derivative of Marquardt’s Percent Standard Deviation (MPSD) (Mahmoud et.al., 2017; Xu rt. al., 2013):
Isotherm models statistical for adsorbing Pb(II) by GOSCh adsorbent.
|
Models |
Parameters |
Pb(II) |
|
Freundlich |
R2 |
0.99 |
|
MPSD |
0.33 |
|
|
Langmuir |
R2 |
0.93 |
|
MPSD |
0.63 |
|
|
Harkins-Jura |
R2 |
0.75 |
|
MPSD |
6.99 |
|
|
Duldinin-Redushkevich |
R2 |
0.79 |
|
MPSD |
2.77 |
|
|
Temkin |
R2 |
0.94 |
|
MPSD |
3.26 |
Kinetic model statistical obtained for the adsorption of Pb(II) using GOSCh.
|
Models |
Parameters |
Pb(II) |
|
Pseudo-first-order
|
R2 |
0.17 |
|
Pseudo-second-order
|
R2 |
0.99 |
|
MPSD |
0.0033 |
|
|
Elovich |
R2 |
0.89 |
|
MPSD |
7.52 |
|
|
Intra-particle |
R2 |
0.73 |
|
MPSD |
5.57 |
Amendment: Page 10, line 296-302 and Table 2; Page 12, Table 3
For Point 7, the authors didn’t understand the question and didn’t carry out statistical analysis for Figures 5, 6, 7, 11, and 12. The MPSD values are not required since R2 values are presented.

Reviewer 2 Report
Authors claim that the hydrophilicity could be enhanced by silica, but they did not measure it. This must be measured before reconsidering the paper.
The paper is full of assumptions. Authors must show this with data.
The revised manuscript still has many typos.
What is the difference between GO-SiO and GO-Si, and GOSCh and GOCh?
Where is Figure 13?